# The Benefits of emotion Regulation Interventions in Virtual Reality for the Improvement of Wellbeing in Adults and Older Adults: A Systematic Review

**DOI:** 10.3390/jcm9020500

**Published:** 2020-02-12

**Authors:** Jessica Isbely Montana, Marta Matamala-Gomez, Marta Maisto, Petar Aleksandrov Mavrodiev, Cesare Massimo Cavalera, Barbara Diana, Fabrizia Mantovani, Olivia Realdon

**Affiliations:** 1Department of Human Sciences for Education, University of Milano-Bicocca, Piazza dell’Ateneo Nuovo 1, 20126 Milano, Italy; marta.matamalagomez@unimib.it (M.M.-G.); marta.maisto@unimib.it (M.M.); p.mavrodiev@campus.unimib.it (P.A.M.); barbara.diana@unimib.it (B.D.); fabrizia.mantovani@unimib.it (F.M.); olivia.realdon@unimib.it (O.R.); 2Department of Psychology, Catholic University of the Sacred Heart, Largo Gemelli 1, 20100 Milan, Italy; cesarem.cavalera@unicatt.it

**Keywords:** emotion regulation, treatment, wellbeing intervention, adults, virtual reality, systematic review

## Abstract

The impact of emotion regulation interventions on wellbeing has been extensively documented in literature, although only in recent years virtual reality (VR) technologies have been incorporated in the design of such interventions, in both clinical and non-clinical settings. A systematic search, following the Preferred Reporting Items for Systematic Reviews and Meta-Analysis (PRISMA) guidelines, was therefore carried out to explore the state of the art in emotion regulation interventions for wellbeing using virtual reality. The literature on this topic was queried, 414 papers were screened, and 11 studies were included, covering adults and older adults. Our findings offer an overview of the current use of VR technologies for the enhancement of emotion regulation (ER) and wellbeing. The results are promising and suggest that VR-based emotion regulation training can facilitate the promotion of wellbeing. An overview of VR-based training interventions is crucial for better understanding how to use these tools in the clinical settings. This review offers a critical debate on the structure of such intervention protocols. It also analyzes and highlights the crucial role played by the selection of the objective and subjective wellbeing assessment measures of said intervention protocols.

## 1. Introduction

Emotion is a cultural and psychobiological adaptation mechanism that allows each individual to react flexibly and dynamically to environmental contingencies [1]. Emotions give meaning to our lives, intensify our connection with others, inform us about our needs and feelings and motivate us to make changes [2]. Emotions are multidimensional phenomena. A single emotion comprises of: a cognitive appraisal, a physical sensation, an intention, a subjective “feeling”, a motor response and, in most cases, an interpersonal component [1,3]. Emotion Regulation (ER) denotes a set of mental processes that influences which emotions we have when we have them and how we experience and express them [4,5]. It is a dynamic process inherent to the mental functioning of human beings, aimed at down or up-regulating positive or negative emotions in order to reach desirable states [6,7]. Emotion dysregulation denotes the undesired intensification or deactivation given by the person’s inability to manage or process emotions effectively [8,9]. Hence, dealing with emotionally rich experience is part of emotional regulation. A priori assumptions as to whether any particular form of emotion regulation is necessarily good or bad do not exist [10]. This is important because it aims to avoid a type of distinction that is made, for example, between coping strategies, as more or less adaptive regardless of the context [11,12]. The emotion regulation process is a mechanism may be used to make things either better or worse, depending on the context. Furthermore, in line with a functionalist perspective, regulatory strategies may accomplish desired goals but still be perceived by others as maladaptive [13], such as when a child cries loudly in order to get attention [7]. A notable contextually adaptive ER strategies is reappraisal. It changes the way one thinks about a potential emotion-eliciting event. Another one is suppression, which changes the behavioral response to an emotion-eliciting event [14]. Thus, an effective situation-based regulation of emotions is necessary for permanent and enduring change in a person’s growth and their social functioning [14] and subjective wellbeing [15,16,17].

The growing development of new technologies and the interest in applying them in the field of psychology have led to the development of novel virtual reality (VR) systems for neuro-rehabilitation [18], or the treatment of different mental health disorders [19,20,21,22] with the aim of generating an engaging and realistic virtual world in line with the needs of the person [23]. Applying virtual reality in psychology has one major advantage. It allows researchers and clinicians to create life-like experiences in a safe environment such as a laboratory or a clinical setting [24,25]. VR-based assessment and treatment allows to keep complex variables under control while preserving the complexity of real-life experiences [26]. In this regard, the use of virtual reality is promising, because it allows real-time measurement of cognitive, emotional, physiological, and behavioral responses in a variety of “real-life” situations while allow for full experimental control [27,28]. In particular, concerning the application of VR in clinical psychology, virtual environments have been widely used to enhance the use of successful ER strategies [29]. Since we know that effective ER strategies have led to several important outcomes regarding mental health, subjective and psychological wellbeing, and relationship satisfaction [30]. Hence, positive changes in emotion regulation are an important outcome in the mental health interventions and the development of new technologies such as VR systems could facilitate and increase the positive outcomes of such ER strategies [29,31]. It has been demonstrated that VR systems can evoke emotional experiences that lead to psychologically valuable changes through an enhanced sense of presence in a virtual environment [32,33,34]. Hence, it is possible to create a sense of ‘being there’ in a virtual world by designing highly immersive VR experiences, which rely on multisensory feedback mechanisms [35]. 

Health and wellbeing are considered as indispensable resources for societies and human development [36]. The World Health Organization (WHO) has placed wellbeing on the “Health 2020: the European policy for health and well-being” agenda as an objective for social progress [37]. Following the WHO, the purpose of the present review is to investigate whether wellbeing can be enhanced using new technologies, such as virtual reality. Specifically, the present systematic review aims to better understand the efficacy of emotion regulation interventions for wellbeing, by using virtual reality systems in adults and older adults without psychopathological conditions. 

## 2. Method

A systematic review of the scientific literature has been performed to identify studies that reported VR-based ER interventions for wellbeing in healthy and clinical adults and examined the structure of their protocols. The methodology is presented in the following paragraphs. 

### 2.1. Search Methodology

Preferred Reporting Items for Systematic Reviews and Meta-Analysis (PRISMA) guidelines were followed [38]. Four high-profile databases (PubMed, Embase, Scopus, and Web of Science) were used to perform the computer-based research on the 30th of September 2019 (see Table 1 and Figure 1). According to the PICO format, we defined the review question as, “is VR training for emotion regulation, compared to treatment as usual, effective in improving wellbeing in adults (with psychological distress).” We then proceeded with the definition of keywords for the search strategy. The string used to carry out the search strategy was (“virtual reality” OR “virtual environment*” OR “digital intervention*” OR “digital technologies”) AND (“emotion regulation” OR “affect regulation” OR “wellbeing”). The initial searches on the databases yielded 530 results. Duplicates were removed leaving 414 articles for further evaluation. Table 1 shows the details of the results for each keyword on each database used. 

### 2.2. Study Selection and Inclusion Criteria

This systematic review aims to evaluate the wellbeing and emotion regulation outcome of VR-based interventions in adults and older adults without psychopathological conditions. Given that the interest in VR continues to grow, researchers must focus on how the characteristics of VR systems and the different aspects of the training tasks could influence the intervention outcomes. The aim of this review is to provide knowledge and guide researchers in the selection of the most appropriate VR experience for ER interventions. The flow chart of the search strategy results, according to the PRISMA flow diagram, is shown in Figure 1.

The present systematic review considered randomized control trials, nonrandomized control trials, intervention studies, and case-control studies. Studies on emotion regulation for wellbeing with virtual reality (VR) devices in healthy or pathological adults and older adults presenting the following clinical conditions: traumatic brain injury, motor disabilities, tumor, chronic conditions (heart failure and chronic pain), were included. The review only includes studies in the English language, and which satisfied strict criteria for eligibility (research studies, interventions for adults and older adults, VR non-/semi-/immersive and immersive interventions, interventions for emotion regulation, interventions for wellbeing, healthy population and clinical patients but not psychopathological, wellbeing outcomes). Articles that treat psychopathological disorders such as post-traumatic stress disorder, phobias, substance abuse or psychosis, or lacked necessary information for review in the full-text or the abstract were excluded. Reviews, meeting abstracts, proceedings, poster presentations, notes, case reports, letters to the editor, assessment protocols, editorials, and other editorial materials were also excluded. Retrospective studies were not included because the area of interest requires post-intervention outcomes.

### 2.3. Risk of Bias Assessment

To assess the risk of bias, the reviewers followed the methods recommended by The Cochrane Collaboration Risk of Bias Tool [39] and the STROBE Statement [40]. Three reviewers (J.I.M., M.M.-G., and M.M.) independently assessed the risk of bias of each included study against key criteria: random sequence generation, allocation concealment, blinding of participants, personnel, and outcomes, incomplete outcome data, selective outcome reporting, and other sources of bias. The following judgments were used: low risk, high risk, or unclear (either lack of information or uncertainty over the potential for bias). Disagreements were resolved through consensus, and another author was consulted to resolve disagreements if necessary. In particular, the selected studies followed strict criteria in the methods, including presenting critical elements of study design, clearly defining all outcomes, describing the setting and relevant dates, including periods of recruitment and exposure, giving sources of data and details of methods of assessment (measurement).

## 3. Results

Of 414 non-duplicate studies, 386 did not fit the preliminary inclusion criteria; specifically, they did not present ER interventions for wellbeing using VR systems in adult and older adult populations. Subsequently, the full text of 28 articles was retrieved and the studies were evaluated for the specific inclusion criteria. Of 28 studies, only 11 passed the full-text screening phase, while 17 studies were excluded for the reasons that follow: Not interventions (= 4); Results not reported (= 7); Qualitative/descriptive study (= 6). 

### 3.1. Flow Chart of the Results

The present flow chart (Figure 1) shows a summary of the research strategy (presented previously in Table 1), the methodology followed during the study selection process, and the final included studies according to PRISMA Guidelines.

### 3.2. Risk of Bias 

The majority of the studies except one [41] exhibited a medium and high risk of bias across multiple dimensions. Table 2 shows the results for the risk of bias assessment. All the studies included in this review reported the sampling method [41,42,43,44,45,46,47,48,49,50,51], although the performance and the detection biases during blinding phase were unclear for all but one [41]. Concerning the outcomes, only two studies [45,47] presented high risk of bias for missing data handled appropriately or for missing a match between methods and results. Among other risks, we reported a high risk of bias for a small sample size with a range from eight to fifteen participants in three studies [44,47,51]. We considered important to report also a high risk of bias for lacking a control group for an experimental comparison [44,45,47,48,49,50,51]. Lastly, only one study addressed to patients have reported a high risk for no homogeneous clinical sample due to differences in clinical diseases and their specific characteristics [44] that might affect the interpretation of the outcomes. A clean sample is crucial for the comprehension of the ramification of disease on emotional functioning.

### 3.3. Study Characteristics

Table 3 shows the studies’ characteristics according to extraction parameters. Eleven studies were analyzed to understand the usefulness of interventions for emotion regulation and wellbeing using virtual reality (VR) systems. In order to accomplish the aims of the systematic review and to facilitate the understanding of the selected studies, the following clusters in Table 3 were considered: (1) Authors; (2) Year; (3) Sample (N); (4) Sample characteristics; (5) Mean age (SD or range); (6) VR Task; (7) VR Set-Up; (8) emotion regulation and/or wellbeing assessment; (9) Primary Outcomes.

### 3.4. Interventions for Adults and Older Adults

#### 3.4.1. Age Differences in Emotional Experience

Several investigations suggest that VR-based ER interventions in adults and older adults could improve quality of life, physical and mental health and delay the onset of health disorders [29,30]. We considered it appropriate to divide the discussion of the results according to the age of the participants. This choice was based on two reasons. Firstly, studies in the adult and elderly populations have different objectives. Secondly, emotional experience, expression and regulation, like all psychological phenomena, depend on physiological functioning [52]. In regard to the former, Gross et al. examine age differences in participants’ reactions to negative events, showing that older people report better control over emotions compared to younger people [53]. In regard to the latter, studies observe how heart rate increases and epithelial cells lining the vasculature either constrict or dilate in response to an arousing stimulus. This overall pattern of reactivity is reduced among older adults [54,55]. 

In the following paragraphs we discuss the characteristics of the selected VR interventions. The examined studies focus on the use of virtual environments for intervening on emotion regulation processes, and for improving the wellbeing of healthy and clinical populations.

#### 3.4.2. Interventions for Adults 

The evaluated interventions for adults and older adults (specifically the VR characteristics, the VE content, and the aims) are summarized in Table 4 and Table 5 respectively. 

#### 3.4.3. Virtual Environments (VE) for Healthy Participants

In what follows we examine studies conducted in a healthy population (see Table 4). Villani and Riva’s used two virtual environments, one depicting a waterfall zone, and another an island, to induce relaxation and to enhance the wellbeing of participants [42]. The authors employed these scenarios in a way that maximizes the sense of presence in the virtual world, which enhances the quality of the relaxation experience [42]. Konrad et al. [41] used a web-based technology-mediated reflection (TMR) application, called the “Mood Adaptor,” to enhance ER. It is a systematic process that reviews rich digital records of past personal experiences. According to the authors, such autobiographical memory approach increases general wellbeing. In their intervention, participants were instructed to write down a positive thought while experiencing a negative mood and, vice versa, write down a negative thought while in a positive mood [41]. Their explanation relies on the fact that recalling negative experiences while in a positive mood can update the emotional appraisal of the past experience and generate more adaptive perspectives of the past [41]. 

Weerdmeester et al. [48] have examined the role of self-efficacy in the context of biofeedback video games for ER. A pilot study was conducted with a VR videogame, called DEEP, which uses respiratory-based biofeedback to help individuals cope with stress and anxiety. Self-efficacy was found to be a significant predictor of physiological regulation, and a key factor in the improvement of mental wellbeing [48]. The biofeedback paradigm has been defined as a system in which physiological activity is continuously measured and fed back to the user in real-time [48]. In fact, in Weerdmeester’s study, deep, calm breathing allows the player to stay afloat and move smoothly through the underwater world [48]. Lorenzetti et al. [51] have developed a neurofeedback game using a virtual environment as a medium to convey a real-time sensory feedback to participants, in association with ongoing tenderness, anguish, and neutral emotional states. Lorenzetti et al., used a BCI-based neurofeedback system in which neural activity was linked to the color of the virtual environment which allowed the real-time visualization of the fluctuation of emotional states. Orange denoted tenderness, purple—anguish, and natural light tones—neutral disposition [51]. Participants were instructed to experience tenderness or anguish as intensely as possible in the respective trials and to volitionally increase the intensity of their emotions [51]. 

Bornioli et al. [49,50] support the large amount of evidence concerning the benefits of walking in natural areas. In their two studies, participants had to walk, in five different, virtually recreate, sites: a pedestrianized historic environment in Bristol’s Old Town, characterized by neoclassical buildings and cobbled paving; a pedestrianized modern environment in a complex of concrete and glass-fronted buildings; a pedestrianized environment with a mix of greenery and historic elements, framed by the Bristol Cathedral; a commercial road with high street retail outlets and cafés and a single-lane road with moderate moving traffic, constituted by cars and buses; and an urban park. The outcomes underline the crucial features that make walking positive for psychological wellbeing and encourage this activity [49,50].

#### 3.4.4. Virtual Environments for Patients

Other studies have used interventions on patients with specific pathologies (see Table 4) using virtual environments to enhance ER and wellbeing. Two of these used Mindfulness-based Stress Reduction (MBSR) Meditation through virtual environments as a non-pharmacological approach to treating chronic disorders, such as chronic pain [43], and patients with traumatic brain injuries [44]. In both studies, virtual meditation was significantly more effective than MBSR alone in reducing reported levels of pain [43], and at making participants experience a higher level of satisfaction [44]. Singh et al. have examined the impact of VR games on psychological wellbeing, upper limb motor function and reaction time in adults with physical disabilities. In the latter study, wellbeing was a secondary outcome, as a positive consequence of the physical reactivation of the patient through the use of motor task games [47]. 

#### 3.4.5. Virtual Reality Set-Up

The examined studies have used different types of virtual reality systems. Among the studies that have used immersive VR systems, two have selected a head-mounted display in combination with a smartphone and a head-tracker [42,44]. However, in the study conducted by Tong et al. [43], the authors used a Virtual Meditative Walk system that requires the use of a stereoscopic VR display. The display was mounted on a movable arm to ensure flexibility and to ease patient comfort. Furthermore, the authors used a sensor in order to track changing arousal levels, which are small clips put onto two of the patient’s fingertips [43]. In the DEEP biofeedback game study, participants had to use deep diaphragmatic breathing in order to navigate in an immersive virtual environment through an enchanted underwater world [48]. The game used a customized controller belt that measures the expansion of the player’s diaphragm and a head-mounted device. Other studies have used non-immersive devices, such as mobile applications [41]; Nintendo^®^ Wii Fit [47]; tablet and laptop [45,46,49,50]; and computer-based interface (BCI) [51]. 

#### 3.4.6. VR Interventions for Older Adults

Positive emotions have a significant influence on mental and physical health [56,57]. Their role in the wellbeing of the elderly has been established in numerous studies, so it is worthwhile exploring how older adults can improve the number of positive experiences in their daily lives [58]. Digital technologies are a powerful tool to enhance social inclusion [59], to support a more active and independent life in older adults and consequently to facilitate their wellbeing [60]. Among the presently investigated studies (see Table 5), Hasan et al. [45] and Baez et al. [46] proposed VR interventions focused on enhancing abilities for the maintenance of autonomy in older adults. Hasan et al. have carried out a two-year project on the social use of information and communications technologies (ICT) in older adults. The interventions consisted in the establishment of computer kiosks in aged-care facilities and weekly classes for developing ICT skills and enhancing wellbeing [45]. The study conducted by Baez et al. was a home-based intervention program to promote physical activity in older adults [46]. In this study, a web and a tablet application have been delivered to participants in order to enable and motivate them to participate in a home-based group training session, under the supervision of a human coach. After a period of eight weeks of training, participants presented a significant improvement of subjective wellbeing [46]. Moreover, in two studies conducted by Bornioli’s and co-authors with older adults, the authors used a virtual walk intervention to enhance physical activity intentions and psychological wellbeing [49,50].

### 3.5. Outcome Measures for Wellbeing

#### 3.5.1. Physiological Markers

A large number of studies among those evaluated in this systematic review use physiological markers as an objective measure to assess the effectiveness of VR interventions for ER. In this regard, the appraisal theory of Scherer allows us to explain the link between emotional states experienced by participants during interventions and their physiological responses [61,62]. In particular, this occurs due to the direct connections between stimulus evaluation check units and response modalities in the neuroendocrine system, autonomic nervous system, and somatic nervous system, independently from action tendencies [61]. A study by Villani and Riva (2008) [42] shows a significant reduction of anxiety, as well as a significant improvement of positive emotional states, in particular, relaxation, measured through physiological parameters that demonstrate good fluctuations of respiration rate, heart rate, and skin conductance. Tong et al. use a VR system that incorporates biofeedback mechanisms to support the learning of mindfulness practice [43]. This technological intervention may be an effective and long term non-pharmacological alternative to traditional pain management [43]. Furthermore, a biofeedback system has been used through the implementation of a belt for tracking the breathing rhythm of the participant during a VR relaxation-based game [48]. Biofeedback is an interesting mind-body therapy using electronic instruments to help individuals gain awareness and control over physiological processes [63]. Biofeedback is the process of measuring an individual’s physiological activity such as brain activity, heart rate or breathing, and subsequently provide real-time information about this activity to the individual [64]. Through this feedback component, participants become more aware of their physiological activity, so they can learn how to gain control over it and improve their wellbeing [65,66]. 

In addition to the studies examined above, a neurofeedback (NFB) system had been used to enable the participants to regulate their brain activity, to enhance and recover emotional and cognitive capacity and to improve their underlying neurobiology [51]. Neurofeedback makes use of electroencephalography (EEG) biofeedback to guide participants in modifying their cortical activity, alter their states of consciousness, and affect cortically mediated physical and psychological functioning [67]. Novel forms of NFB, such as ones based on real-time functional magnetic resonance imaging (rtfMRI), hold a still vastly unexplored potential for complex technological applications, such as the currently discussed VR-based ER interventions [68].

#### 3.5.2. Wellbeing Scales Outcomes

Given the primary objective of the present manuscript, in Table 6 we summarized an overview of the main wellbeing measures used in the evaluated studies. Several wellbeing measures were used as pre- and post-intervention measures: the State and Trait Anxiety Inventory (STAI) to evaluate anxiety levels [42,51]; the Beck Depression Inventory (BDI) to assess depression levels [51]; the Depressive, Anxiety and Stress Scales (DASS) to assess state anxiety and depression levels [47]; the Positive And Negative Affect Scale (PANAS) to measure the positive and negative affect through 10 positive and 10 negative moods/emotion adjectives [42,51]; the Coping Orientation to Problems Experienced Questionnaire (COPE) assessed the different strategies commonly activated in daily problem solving [42]. Results show a significant reduction of anxiety [42,47,51], and a significant improvement of positive emotional states [42,51]. Given that there is no universal measure of wellbeing, Konrad et al. include both hedonic (e.g., pleasure, satisfaction), and eudaimonic (e.g., meaning, personal growth) scales to triangulate different measurement perspectives: Subjective Happiness Scale, Satisfaction With Life Scale, and Ryff Scales of Psychological Wellbeing [41]. Moreover, positive intervention outcomes have been measured in both clinical and healthy populations with the Mindfulness Attention Awareness Scale (MASS), and the Satisfaction With Life Scale (SWLS) [44]. Tong et al., did not use a scale to monitor changes in mindfulness awareness, rather, they used the pain level and biofeedback data as an objective wellbeing measure [43]. However, there are no subjective wellbeing measures in that study [43]. Another interesting measure that has been used in the reviewed studies is the wellbeing scale of the Multidimensional Personality Questionnaire (MPQ) that underlined improvements in both groups with no significant difference between groups in social wellbeing outcomes [46].

The Social Care-Related Quality of Life (SCRQoL) Scale was administered to track changes in the quality of life of older users to their daily routines, based on the outcome domains of social care-related quality of life identified in the Adult Social Care Outcomes Toolkit (ASCOT) [45]. Moreover, Weerdmester and colleagues selected the Trier Social Stress Test, a tool for investigating psychobiological stress responses [48]. Pre- and post-wellbeing assessments used the stress and hedonic tone measures based on the University of Wales Institute of Science and Technology Mood Adjective Checklist scale (UWIST MACL), to measure different stress-related states (nervous, tense, relaxed, calm), and the hedonic tone (happy, content, sad, sorry) [49,50]. 

## 4. Discussion

This review aimed to investigate how emotion regulation interventions using virtual reality (VR) systems can enhance the wellbeing in healthy and clinical adults and older adults, without presenting psychopathological conditions. In recent years there is a growing interest in the use of advanced technologies in supporting wellbeing and promoting health [34], following emerging evidence that technology can increase emotional, psychological and social wellbeing [69]. Technological approaches appear to be more advantageous in terms of intensity and duration of treatment, costs, and usefulness in the continuity-of-care [70,71]. Considering the multifaceted construct of wellbeing, in what follows, we strive to unpack the complex picture provided by the results of the hereby examined studies. This complexity emerges from the different functional aims of the studies, employed in pursuit of wellbeing enhancement. Three studies (27%) were developed for relaxation or following the Mindfulness-Based Stress Reduction (MBSR) protocol [42,43,44]; two studies (18%) used biofeedback or neurofeedback as peripherical technique for the regulation of physiological arousal [48,51]; one study (9%) used survey forms for a mood regulation and improvement wellbeing [41]; four studies (37%) intended to encourage a participants’ behavioral and physical activation in order to enhance their subjective wellbeing through outdoor [49,50] or indoor activities [46,47]; one study (9%) aimed to enhance ICT-skills of elderly people in support of their social functioning and wellbeing [45]. 

In the examined studies, virtual reality has been used in different ways to promote wellbeing in healthy and clinical populations. Some authors used mindfulness as a non-pharmacological approach to manage participant’s emotion regulation abilities and to enhance their wellbeing [43,44]. It has been demonstrated that immersive VR can be used as a powerful pain control technique to manage and modulate pain in healthy and clinical populations [72,73,74,75]. This is in line with the study conducted by Tong et al., in which the authors found that the use of an immersive VR combined with a biofeedback system, can be a helpful approach for managing chronic pain on a long-term scale [43]. 

In the present review, the combination of VR with biofeedback systems was particularly interesting. In recent years the development and use of game-based biofeedback to promote physical and mental wellbeing is growing [76,77,78]. An example, is the study of Weerdmester et al., in which an intervention based on the VR game “Deep”, combined with a biofeedback system, highlighted positive and promising outcomes to help individuals cope with stress and anxiety [48]. This technique relies on visual and auditory feedback to guide participants in becoming more aware of their breathing and incentivizes the adoption of a more calm and relaxed breathing pattern [66,79,80]. Furthermore, promising results of game-based biofeedback have been found for emotion regulation and the treatment of stress and anxiety [81]. 

Neurofeedback (NFB) is a type of biofeedback which facilitates the real-time voluntary regulation of brain activity through a brain-computer interface [67,68]. Lorenzetti et al. [51] showed that NFB has promising effects on enhancing behavior, cognitive and emotional processes in healthy subjects. Performing exercises in an immersive VR environment is shown to decrease depression, anxiety, and stress [82]. Intervention outcomes, are in accordance with the literature, highlight a reduction in anxiety scales [42,47,51], and an improvement of positive emotional states [41,42,44,49,50,51]. 

For instance, Singh et al. [47] suggest that interactive VR games can be used as an exercise tool to improve psychological wellbeing and reaction time among adults with physical disabilities, for whom the use of technology may promote adherence, motivation, and participation in physical activity and exercise programs [46,47,49,50]. Positive results can also be seen in older adults, as demonstrated in the intervention program proposed by Baez et al. [46]. This intervention program was designed to promote physical activity in older adults, either in a group or in a home-based setting, showing equivalent health outcomes for both groups, but different results when considering adherence [46]. These interventions point out the importance of immediate wellbeing responses, in that, positive affect can be beneficial to long-term health [57,83]. A large number of studies have demonstrated the association of regular physical activity with positive outcomes for improving health and wellbeing conditions [84,85]. Moreover, home-based intervention programs aimed at promoting physical activity in older adults, either in a group or individually, have demonstrated the potential to improve health and functional performance [86]. The social wellbeing in the elderly is another issue that requires attention. Hasan et al. identified the complexity of this problem domain and engaged older adults through activities aimed at enhancing their ability to remain productive [45]. 

These results are an important outcome for wellbeing. The literature demonstrates that VR-based interventions are as effective as traditional ones in the treatment of different mental disorders, but can achieve positive outcomes in less time [87]. The difference between VR-based interventions and treatment as usual (TAU) remains to be clarified. Some of the currently examined studies, highlights a significative difference between the experimental and the control group [43,46], but further randomized controlled trials are required to achieve a better understanding of the effectiveness of VR compared to non-VR interventions.

Why are VR-based ER interventions important for wellbeing? To address this question, we have to consider several things. Emotions can be transient phenomena which emerge from momentary situational goals and subside with the short-term achievement of such goals [88]. Emotions can also derive from long-term goals and enduring values concerning health, close relationships, and important work-related projects [89]. Hence, knowing the specific antecedents of specific emotions is crucial to better understand the complexity and potential of technological ER interventions. 

The present review underlines that virtual reality systems evoke a general positive emotion and could promote a healthy life [90] and an optimal state of functioning such as flow [91]. The sense of flow in VR is evoked when the user is immersed in a highly rewarding activity, accompanied by a high sense of control [92,93]. Flow can subsequently promote the feeling of immersion in the virtual environment [94]. Four studies among those investigated, have used an immersive VR system and have emphasized the importance of the immersion degree in a virtual environment. These studies used a head-mounted display (HMD) through which participants can be immersed in an interactive virtual reality scenario [19]. Through the HMD, the experimenter may provide different sensory input, as well as synchronize participants’ movements with the generated virtual feedback (e.g., avatar movements, or reaching virtual objects) [20]. In order to increase the sense of immersion, feedback across different sensory channels is provided, such as, visual, acoustic, and tactile. This can also be achieved through the use of input tools such as trackers, gloves and other controllers, that allow to continuously monitor the position and movements of the users, and synchronize them with the VR interactions [32,95]. From a psychological point of view, at the basis of the feeling of immersion in VR is the sense of “presence,” defined as the psychological sensation of “being there” in the virtual scene instead of in the physical and real environment [28,35,96]. The sense of presence has also been defined as the “feeling of being in a world that exists outside of self” [93,94]. The sense of presence in a virtual environment is given not only by the realism of its graphics but also by subjective characteristics, such as the potential of a given virtual scenario to elicit certain emotional responses [42]. In this way, a VR-based intervention can modify personal experiences by inspiring users to try new things [97] and allow them to modify habitual emotional responses to specific situations [7,98]. A VR system has the potential for a laboratory vs. everyday functioning rapprochement. Virtual environments allow to immerse participants in digitally recreated real-world activities which can be enacted in the safety of the laboratory setting [99]. The ability of VR systems to reproduce the complexity of real-life situations is a peculiar element for ER interventions. Due to this peculiarity, the user can be immersed in a complex virtual environment that requires the use of a complex set of ER strategies that form a dynamic pattern. An ER strategy does not have to be unique and universal, but in order to be useful and transferable to daily life, it must be adaptive and generalizable across situations [6,100]. In this sense, the use of virtual reality is promising, because it allows the user to learn complex ER strategies and, potentially, experience them in different environments similar to the real world. Lastly, a VR system provides experimental control and dynamic presentation of stimuli in ecologically valid scenarios [18,26,27,71,101]. This can be done by measuring the real-time cognitive, emotional, physiological and behavioural responses in a variety of life-like virtual situations [28]. 

The obtained results show that interventions using virtual reality systems allow people to change or improve their ER strategies. One such strategy is “situation selection” [4]. It is an antecedent-focused strategy that examines all the actions that we execute before the emotional response has become fully active and has modified behavioral or physiological responses and selects the best action [55]. Furthermore, technology can provide different options to appraise emotional stimuli, hence different cognitive appraisals pertaining to the same potential trigger and eventually different emotional responses [7]. In regard to patients with specific disorders, VR interventions can facilitate the approach to rehabilitation therapy by training more contextually adaptive ER strategies such as “reappraisal”, which lead to better short-term affective, cognitive, and social consequences, compared to less adaptive strategies such as “suppression” [14]. VR systems improve patients ability to regulate the emotions that accompany their everyday experiences with their health condition [29]. Such targeted interventions that facilitate a reappraisal, can elicit a subsequent “cognitive change” [7,55]. This change refers to selecting what meaning people attach to the specific intervention [4,7]. External variables that are properties of the stimulus can influence the choice of emotion regulation strategy; in fact, reappraisal affordances, defined as the opportunities for re-interpretating a stimulus, which are inherent to the stimulus itself, can greatly shape such choices [13]. The personal meaning assigned to a specific situation is crucial for establishing habitual experiential, behavioral and physiological responses which repeat in that same situation [5]. 

One of the main aims of emotion regulation interventions is to modify the emotional responses of the subjects to one specific situation [102]. The outcome of the interventions reviewed in this article could be interpreted according to “The process model of emotion regulation” of Gross. According to Gross reappraisal-like processes could influence emotional responses. Efforts to down-regulate emotion through reappraisal alters the trajectory of the entire emotional response, leading to lesser experiential, behavioral, and physiological responses [7,102]. An example of this process is found in the studies that used Mindfulness-Based Stress Reduction (MBSR) protocol in virtual reality to minimize the negative emotional impact in patients with traumatic brain injuries [44] and the Meditative Virtual Walk for patients with chronic pain [43]. Mindfulness is a core skill of Dialectical Behavioral Therapy employed in the treatment of emotion dysregulation. It consists in observing, describing, and “allowing” emotions to flow without judging them or trying to inhibit them. Mindfulness is hypothesized to influence the habitual or automatic response to emotional behaviors and their associated appraisals. Hence, in comparison to Gross’s Model, mindfulness may alter automatic response tendencies by altering the habitual approach or avoidance response to that of a non-judgemental awareness of the emerging emotions [103]. During the interventions the individual can reappraise the emotional responses to their health conditions. This type of reappraisal can also be seen in the intervention conducted by Konrad et al. [41]. In this intervention, subjects were asked to describe a negatively appraised past experience while in a positive mood. This was proven to update their emotional response to the past experience, encouraging the emergence of more adaptive perspectives [104,105]. Finally, virtual reality tools promote the sense of environmental mastery and the continued development of competence and self-knowledge, both in clinical and non-clinical populations [106,107]. However, there is a major difference between both populations. In a healthy population, an ER intervention can cultivate more adaptive, for daily life, ER strategies. In a clinical population, ER interventions are useful for increasing the patient’s ability to handle their specific pathology. 

In conclusion, the choice of wellbeing measures is essential to assessing the efficacy of the intervention. Humans do not act mechanically, but rather according to their subjective interpretation of the world. Hence, objective indicators alone may not be sufficient to hedge the several conceptualizations of wellbeing [108]. It is fundamental to gain knowledge about the subjective interpretations of emotional states. This can happen by directly asking people about their emotions, perceptions, and evaluations [108]. Currently, wellbeing is not a clearly defined concept and the present review shows that several ways to measure wellbeing exist [36]. Despite that difficulty, it is important for studies to clarify the theoretical framework of wellbeing in which the intervention intends to operate. The choice of the theoretical framework can better guide the design of the intervention and the choice of more adequate wellbeing measures.

## 5. Conclusions

In conclusion, the results of this systematic review show that technology can improve the ability of people to handle emotionally-rich life situations by training more contextually adaptive emotion regulation strategies.

The use of virtual reality in this sense is promising because it allows the user to learn complex ER strategies in the context of life-like digital environments. VR interventions can modify the user’s ER by inspiring new actions, allowing for the modification of the emotional response across a reappraisal of emotional stimuli, and subsequently, memorizing the re-evaluated experience. Finally, virtual reality is a tool that fosters a sense of environmental mastery, and, a feeling of personal growth and autonomy. 

The literature in this field is going in an interesting direction, and we recommend some future steps based on the findings of this systematic review, for future design, implementation, and evaluation of VR-based ER interventions in healthy adults and older adults. The results of this systematic review underly the importance of an appropriate assessment and highlight the positive effects of assessing both subjective and objective measures in future studies to fully evaluate the efficacy of VR interventions for ER. The results from both subjective and objective measures will provide an overall and complete frame of the efficacy of VR interventions for ER. Finally, future research should aim to investigate the underlying mechanisms and factors that may contribute to the effectiveness of biofeedback systems when using VR interventions for ER in order to maximize their positive therapeutic outcomes. 

## Figures and Tables

**Figure 1 jcm-09-00500-f001:**
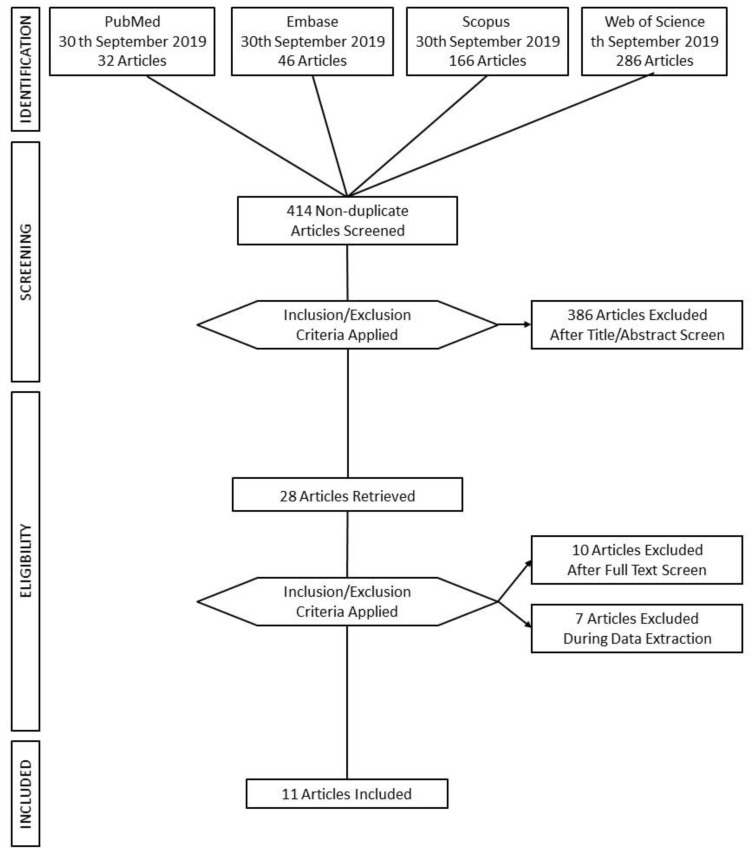
Flow Chart.

**Table 1 jcm-09-00500-t001:** Detailed search strategy.

“virtual reality” OR “virtual environment*” OR “digital intervention*” OR “digital technologies.”						
AND	PubMed	Embase	Scopus	Web of Science		
“emotion regulation”	17	20	40	54		
“affect regulation”	1	1	2	104		
“wellbeing”	14	25	124	128		
**Sub total**	32	46	166	286	**Total**	530
					**Without duplicates**	414

**Table 2 jcm-09-00500-t002:** Risk of bias assessment.

		Random Sequence Generation (Selection Bias)	Allocation Concealment (Selection Bias)	Blinding of Participants and Personal (Performance Bias)	Blinding of Outcome Assessment (Detection Bias)	Incomplete Outcome Data (Attrition Bias)	Selective Reporting (Reporting Bias)	Other Bias
Villani and Riva	2008	low	unclear	unclear	low	low	low	low
Tong et al.	2015	low	unclear	low	unclear	low	low	low
Cikajlo et al.	2016	high	high	unclear	unclear	low	low	high: small sample size/no control group/\no homogeneous clinical sample
Hasan et al.	2016	high	high	unclear	unclear	high	high	high: no control group
Konrad et al.	2016	low	low	low	low	low	low	low
Baez et al.	2017	low	low	unclear	low	low	low	low
Singh et al.	2017	high	high	unclear	unclear	high	high	high: small sample size/no control group
Weerdmeester et al.	2017	high	high	unclear	unclear	low	low	high: no control group
Bornioli et al.	2018	high	high	unclear	unclear	low	low	high: no control group
Lorenzetti et al.	2018	low	low	unclear	unclear	low	low	high: small sample size/no control group
Bornioli et al.	2019	high	high	unclear	unclear	low	low	high: no control group

NOTE: low (risk of bias); unclear (risk of bias); high (risk of bias).

**Table 3 jcm-09-00500-t003:** Study characteristics.

	Authors	Year	Sample (N)	Sample Characteristics	Mean Age (SD or Range)	VR Task	VR Set-Up	Emotion Regulation/Wellbeing Assessment	Primary Outcomes
1	Villani & Riva [42]	2008	60 healthy adults	Experimental Group (EG) for three conditions 45 persons (15 for each condition) Control Group (CG) without treatment 15 persons	Range 21–28 years old	relaxation environment + relaxing narrative	Immersive VR Condition: Sony Glastron PLM S-700 with a head-tracker: Intersense Intertrax2 and Semi-immersive DVD Condition: pc (Fujitsu Siemens AMILO Processor, Pentium 4	State-Trait Anxiety Inventory (STAI) and Positive And Negative Affect Scale (PANAS), Visual Analogue Scale (VAS), Coping Orientation to Problems Experienced Questionnaire (COPE) + Physiological Parameters: Respiration Rate, Respiration amplitude, Heart Rate, Heart Amplitude, Skin Conductance, Electromyography	Results show a significant reduction of anxiety and a significant improvement of positive emotional states—in particular, relaxation—measured through self-report questionnaires in all conditions. Physiological parameters showed some good changes related to respiration rate, heart rate, and skin conductance parameters, but less than expected.
2	Tong et al. [43]	2015	13 patients with chronic pain	EG, 7 patients (3 male, 4 female) CG, 6 patients (3 male, 3 female)	Range 35–55 years old (mean = 49, SD = 8.2)	The Virtual Meditative Walk (VMW) + biofeedback	Immersive environment stereoscopic VR display	Numerical Rating Scale (NRS) for Self-Report Pain Levels (values 0–10)	These findings indicate that the VMW (VR paired with biofeedback for MBSR training) was significantly more effective than MBSR alone at reducing reported pain levels among participants.
3	Cikajlo et al. [44]	2016	8 healthy adults and patients	EG, all participants (of which 4 patients with TBI, one with a brain tumor and 4 workers)	Healthy participants Range 27–40 years old; Patients participants Range 24–48 years old	Mindfulness-Based Stress Reduction VR	Immersive head-mounted display Samsung Gear + Samsung Smartphone S6 and Note4	Mindfulness Attention Awareness Scale (MASS) Satisfaction With Life Scale (SWLS) Mini-Mental State Examination Test (MMSE)	Patients achieved very high level of satisfaction (SWLS) at the end of the study. A slight increase in MASS score is also noticeable. All patients had MMSE score 30, except one; his score was 19 at the beginning, 29 at midterm and 26 at the end of the study.
4	Hasan et al. [45]	2016	27 older adults	Elderly group Seniors group	Older Range 70–98 years old; Seniors Range 50–70 years old	Weekly classes for developing ICT skills in aged-care facilities for 2 years	Non-immersive laptops and tablet	Social Care-Related Quality of Life (SCRQoL)	During the 2-year study, the participants developed various computing capabilities. The use of ICT appears to contribute positively to the wellbeing of the elderly: connection, self-worth/ esteem and personal development, productivity, occupation, self-sufficiency, being in control, and enjoyment.
5	Konrad et al. [41]	2016	128 healthy adults	34 in the Incongruent Negative group, 34 in the Congruent Negative group, 30 in the Incongruent Positive and 30 in the Congruent Positive group	Range 18–62 years old (M = 24.56, SD = 8.87); 91 female and 37 male	MoodAdaptor—a technology-mediated reflection (TMR) application	Non-immersive mobile app	Subjective Happiness Scale, Satisfaction With Life Scale, Ryff Scales of Psychological Well-Being and Personal Emotion Scale Participants.	Autobiographical memory enhances positive mood through well-documented self- enhancement biases. Negative thoughts when in a positive mood reduced current mood, while positive thoughts, when in a negative mood, enhances it. Selecting incongruent memories is useful for mood-regulation and consequently for improving wellbeing.
6	Baez et al. [46]	2017	40 older adults	EG, 20 participants CG, 20 participants	Range 65–87 years old	OTAGO personalized exercise program for fall prevention	Non-immersive tablet-based application (10.1inch Sony Xperia tablet)	Physical Activity Enjoyment Scale (PACES), Wellbeing Scale of the Multidimensional Personality Questionnaire (MPQ), Trans Theoretical Model of Behavior Change (TTM), R-UCLA Loneliness Scale	In virtual group exercising, people with lesser physical skills improve to the level of the more fit participants. These results suggest that: the online group could overcome some of the major issues reported in the literature in terms of the negative effect of group-exercising in the motivation of heterogeneous groups; and it helped reduce the effect of the initial skill level and motivation levels of participants in comparison to the trainees complying with the group norm.
7	Singh et al. [47]	2017	15 patients with motor disabilities	EG	Mean age = 22.7, SD = 4.2	Physical activity task	Non-immersive Nintendo^®^ Wii Fit	Depressive, Anxiety and Stress Scales (DASS)	The results of this study demonstrated that there was a significant difference in psychological well-being and reaction time after intervention using interactive VR games.
8	Weerdmeester et al. [48]	2017	72 healthy adults	EG	Range 18–30 (M = 21.5, SD = 2.7); 31% male, 69% female	DEEP VR a virtual reality biofeedback game	Immersive VR game + biofeedback	Trier Social Stress Test	These results provide a promising outlook for using biofeedback video games such as DEEP to help individuals learn how to regulate their physiological arousal engagingly.
9	Bornioli et al. [49]	2018	269 healthy adults and older adults	EG, all participants see five different environments	Range 18–67 years old (M = 31.69, SD = 13.63; 30.9% male, 69.1% females	exposure urban walking task	Non-immersive laptop	University of Wales Institute of Science and Technology Mood Adjective Checklist (UWIST MACL scale), Russell’s circumplex model of affect, Perceived Restorativeness Scale—Short Version (PRS scale)	This study sets out to investigate the immediate psychological wellbeing benefits of virtual exposure to different urban walking settings. The results suggest that walking in high-quality urban settings can have positive outcomes and highlight the negative role of traffic and the potential benefits of historical elements in the affective walking experience.
10	Lorenzetti et al. [51]	2018	8 healthy adults	EG	Range 23–28 years old	exposure to an autumnal nature environment	Non-Immersive virtual environments in a brain-computer interface (BCI) + rrtfMRI-NFB	Beck Depression Inventory (BDI), Trait Anxiety Inventory (STAI) and Positive And Negative Affect Scale (PANAS), Emotion Regulation Questionnaire (ERQ) and Satisfaction with Life Scale (SLS)	The study provides a novel proof of concept and demonstrates the feasibility of the implementation of rtfMRI-NFB using virtual environment and music to elicit the neural activity and measure the neural correlates of specific, complex emotional states. Real-time up-regulation of tenderness engaged the hypothalamic septum area and other regions previously implicated in positive affiliative emotions (i.e., medial frontal cortex and temporal pole, precuneus).
11	Bornioli et al. [50]	2019	384 healthy adults and older adults	EG, all participants see five different environments	Range 18–67 years old (M = 35.01, SD = 13.89)	exposure urban walking task	Non-immersive laptop	University of Wales Institute of Science and Technology Mood Adjective Checklist (UWIST MACL scale), Russell’s Circumplex Model of Affect, Perceived Restorativeness Scale—Short Version (PRS scale)	Results show the crucial features that make walking positive for psychological wellbeing and encourage walking intentions are perceived safety, comfort, and moderate stimulation.

**Table 4 jcm-09-00500-t004:** Interventions for adults.

Study	Sample (Type of)	VE Characteristics	VE Content	Aim of the VR Task
**Virtual Environments for Healthy Participants**				
Villani and Riva (2008) [42]	Healthy	Immersive	A waterfall and a beach of an island	Relaxation and enhancement of wellbeing
Konrad et al. (2016) [41]	Healthy	Non-immersive	Mood survey	Mood regulation and improvement wellbeing
Weerdmester et al. (2017) [48]	Healthy	Immersive	Underwater world	Regulation of physiological arousal
Bornioli et al. (2018) [49]	Healthy	Non-immersive	Five different pedestrian areas of a town	Enhancement of behavioral activation for wellbeing
Lorenzetti et al. (2018) [51]	Healthy	Non-immersive	A landscape of hills and cornfields	Regulation of physiological arousal
Bornioli et al. (2019) [50]	Healthy	Non-immersive	Five different pedestrian areas of a town	Enhancement of behavioral activation for wellbeing
**Virtual Environments for Patients**				
Tong et al. (2015) [43]	Clinical	Immersive	Walk in the forest	Mindfulness-based stress reduction
Cikajlo et al. (2016) [44]	Clinical	Immersive	A river and a mountain landscape	Mindfulness-based stress reduction
Singh et al. (2017) [47]	Clinical	Non-immersive	Three different sports: tennis, bowling, and boxing	Enhancement of behavioral activation for wellbeing

**Table 5 jcm-09-00500-t005:** Interventions for older adults.

Study	Sample (Type of)	VE Characteristics	VE Content	Aim of the VR Task
Virtual Environments for healthy participants				
Hasan et al. (2016) [45]	Healthy	Non-immersive	Social networks, emails	Enhancement of ICT-skills for the improvement of social functioning and wellbeing
Baez et al. (2017) [46]	Healthy	Non-immersive	A gymnasium	Enhancement of behavioral activation for fall prevention and wellbeing
Bornioli et al. (2018) [49]	Healthy	Non-immersive	Five different pedestrian areas of a town	Enhancement of behavioral activation for wellbeing
Bornioli et al. (2019) [50]	Healthy	Non-immersive	Five different pedestrian areas of a town	Enhancement of behavioral activation for wellbeing

**Table 6 jcm-09-00500-t006:** Wellbeing Measure Scales Description.

Scale	Description of Measure
State-Trait Anxiety Inventory (STAI)	A commonly used measure of trait and state anxiety. It can be used in clinical settings to diagnose anxiety and to distinguish it from depressive syndromes. Form Y, is its most popular version. It has 20 items for assessing trait anxiety and 20 for state anxiety. All items are rated on a 4-point scale (e.g., from “Almost Never” to “Almost Always”). Higher scores indicate greater anxiety.
Positive And Negative Affect Scale (PANAS)	A self-report questionnaire that consists of two 10-item scales to measure both positive and negative affect. Each item is rated on a 5-point scale of 1 (not at all) to 5 (very much).
Coping Orientation to Problems Experienced Questionnaire (COPE)	A self-reported questionnaire developed to assess a broad range of coping responses with a score on a 4-point scale (e.g., from “I usually don’t do this at all” to “I usually do this a lot”). There are two main components to the COPE inventory: problem-focused coping and emotion-focused coping.
Mindfulness Attention Awareness Scale (MASS)	A 15-item scale designed to assess a core characteristic of mindfulness, namely, a receptive state of mind in which attention, informed by a sensitive awareness of what is occurring in the present, simply observes what is taking place.
Satisfaction With Life Scale (SWLS)	A 5-item scale designed to measure global cognitive judgments of one’s life satisfaction (not a measure of either positive or negative affect). Participants indicate how much they agree or disagree with items on a 7-point scale (e.g., from 1 “strongly disagree” to 7 “strongly agree”).
Subjective Happiness Scale	A 4-item self-report measure developed to assess an individual’s overall happiness as measured through self-evaluation on a 7-point Likert-type scale.
Ryff Scales of Psychological Well-Being	A psychometric inventory in which respondents rate statements on a scale of 1 to 6 (e.g., from 1 “strong disagreement” to 6 “strong agreement”). It is based on six factors: autonomy, environmental mastery, personal growth, positive relations with others, purpose in life, and self-acceptance. Higher total scores indicate higher psychological well-being.
Wellbeing scale of the Multidimensional Personality Questionnaire (MPQ)	A personality test meant to measure personality that gives ratings on four broad traits: Positive Emotional Temperament, Negative Emotional Temperament, Constraint, and Absorption. High scorers on the specific wellbeing scale describe: having a cheerful happy disposition; feeling good about themselves; seeing a bright future ahead; being optimists; living interesting, exciting lives; enjoying the things they are doing.
Depressive, Anxiety and Stress Scales (DASS)	A 42-item self report instrument designed to measure the three related negative emotional states of depression, anxiety and tension/stress. The rating is based on a 0-3 point scale (e.g., from 0 “Did not apply to me at all” to 1 “Applied to me very much or most of the time”).
Emotion Regulation Questionnaire (ERQ)	A 10-item scale designed to measure respondents’ tendency to regulate their emotions in two ways: (1) Cognitive Reappraisal and (2) Expressive Suppression. Respondents answer each item on a 7-point Likert-type scale (e.g., from 1 “strongly disagree” to 7 “strongly agree”).
University of Wales Institute of Science and Technology Mood Adjective Checklist (UWIST MACL scale)	A scale that measures self-reported mood in the dimensions of energetic arousal, tense arousal, and hedonic tone.

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
