# Peer review of "The Benefits of emotion Regulation Interventions in Virtual Reality for the Improvement of Wellbeing in Adults and Older Adults: A Systematic Review"

_jcm, 2020, doi:10.3390/jcm9020500_

Round 1

Reviewer 1 Report

This paper is quite important for many area(eg. clinic, psychology, computer science, etc.).

Because this work can give many researcher a good advice in many way.

But, the more references are need to be added.

the author determined that which reference is included or not by appropriate own criterion.

However, I think that the criterion is strict.

As a result main references are included insufficiently(only 11 articles).

Because other many articles refered in many chapters, they can be reorganized.

Also, It is necessary to revise the structure of a sentence.

Same or similar sentences are appear in many chapters.

the sentences are quite redundant.

Author Response

We are very grateful to you for the time and involvement you have shown reviewing the manuscript. As you see, we have revised the manuscript and incorporated the comments and recommendations. We have highlighted the text in all the revised part in our manuscript from the original submission to facilitate the review process.

The comments we received were unpacked below in two parts to facilitate a fluid understanding of the changes made. We hope that the changes have improved the quality of the original manuscript considerably. Below welisted your detailed comments (italics) and our responses (red).

“More references are need to be added. The author determined that which reference is included or not by appropriate own criterion. However, I think that the criterion is strict. As a result main references are included insufficiently(only 11 articles). Because other many articles refered in many chapters, they can be reorganized.”

We are aware of it. Thank you for allowing us to explain this point. We completely agree with the fact that the research carried out focused on 11 studies only and that we followed a strict criterion, but it was a thoughtful choice. We selected a specific topic for this review and the selection criteria were considered according to the aim. Four databases were queried in order to conduct the research as broad as possible, but following the selection study process based on the PRISMA Guidelines we had to analyze only 11 studies. You can see the Flow chart in paragraph “3.1. Flow Chart of the Results” (line 157) that shows a summary of the methodology followed during the study selection process, and the final included studies according to PRISMA Guidelines.

“It is necessary to revise the structure of a sentence. Same or similar sentences are appear in many chapters. The sentences are quite redundant.”

Thank you for this critical suggestion for improving the readability and correctness of the English language of our ms.  Please see the changes made according to your comment in the revised version of the ms. We have also taken steps to make the content more streamlined and flowing, by removing redundant parts.

Thank you for your valuable contribution to our work; we feel that our manuscript is much more improved, further to your reviews, especially in its clarity for readers.

Reviewer 2 Report

This interesting article presents a systematic literature review on the use of VR-based interventions to support well-being. Both supporting and curative interventions are presented. The article follows a conventional structure of introduction - method - results - discussion - conclusion.

There is still room for improvement in some aspects. One of the most important is the need to improve the logical structure. This applies both at chapter level and within the individual sections. At chapter level, the information developed in the introduction should be incorporated into the study. For example, the work of J.J. Gross is presented with good reason. But the influence on the design of the study, e.g. by defining the surveyed characteristics, does not really become clear. For example, what influence does the prominently presented distinction between antecedent-focused strategies and response-focused strategies have on the design of the study? Even in the discussion, topics are dealt with that do not occur in the actual study in this way. The discussion should be primarily based on the results. In its current form, the actual study appears to be a somewhat foreign body between introduction and discussion and some parts of the discussion appear superfluous.

Within the sections, the structure should also be critically reviewed. Examples are paragraphs 63-74 (ER and VR) and 75-91 (VR and intervention in psychology). In the current order they lead to duplication, especially when the second paragraph talks about wellbeing. In the reverse order, the principle "From General to Specific" would be respected.

In the current form of the paper I suggest to describe the content of each chapter in the beginning of the chapter as an orientation for the reader (and follow this description in the chapter strictly)

In some places, a more concrete writing style would be required, e.g. in the literature review of the introduction naming of the authers and some of their concrete results is IMHO helpful to improve the flow of the text.

Section 3 lacks a systematic approach to evaluating the results, as the approach followed is hard to understand, it’s a kind of sequenced information. Alternatively, the approach followed should be better motivated using an introductory section. E.g., it does not become really clear, why a distinction is made between older adults and adults? Ok, there may be good reasons, and these reasons may have been explained in the introduction, but short explanations and introductions would be beneficial for this sections (IMHO). E.g., section 3.7.2 could be enriched by a table showing the measurement instruments used and their characteristics. I suggest rewriting this section using a more systematic approach. For example, IMHO, the most important characteristic of VR experiences are the objectives pursued, and I would suggest presenting the findings of the study in an own, first section. Further, an alignment of the studies found with the Gross-model would be beneficial, as the model is mentioned in the introduction and in the discussion, but not in between.

At the moment, in the paper shows almost no striking results, which invite readers to cite the paper. The paper is a nice read but it is hard remembering concrete results. This could be improved by adding some tables with statistics (e.g. a categorization of the objectives of VR interventions, percentages of VR environment types (e.g. immersive and non-immersive) etc.

A proofreading is strongly recommended. There are problems e.g. with article / no article, coherence of numerus of the subject and the verb with/without s. Only for section 1, formal issues are mentioned in the review, thereafter commenting formal issues has been stopped.

Remarks regarding the content

L109-L110: IMHO, the search string lacks parentheses: "virtual reality" OR "virtual environment*" OR "digital intervention*" OR "digital technologies” AND "emotion regulation" OR "affect regulation" AND "wellbeing.". Precedence of the AND operator is higher than the precedence of the OR-Operator, so that the string evaluates to: "virtual reality" OR "virtual environment*" OR "digital intervention*" OR  ("digital technologies” AND "emotion regulation") OR ("affect regulation" AND "wellbeing.") which is not the intention of the authors. BTW: The full stop within the quotation marks is syntactically wrong. According to Table 1, the search string was: ("virtual reality" OR "virtual environment*" OR "digital intervention*" OR  "digital technologies”) AND ("emotion regulation" OR "affect regulation" *OR* "wellbeing."), i.e. the “and” before wellbeing must be an “or”? L119, L123: What is the difference between VR device and VR equipment – my impression is that there is no difference? Please use a consistent wording. Further, device is in my ears connotated with hardware. However, hardware develops fast and therefore is not that relevant IMHO. Ok, probably it should point to the difference between immersive and not immersive VR devices, but IMHO the expressions chosen are misleading. More important are the VR applications or VR experiences, which have to be characterized in the review IMHO. L166, Figure 1: First box: Please explain? Why is there only the database PubMed referenced? What is the difference between citation and article? I suggest relocating this figure into the vicinity of Table 1, because it explains parts of the methodology. Table 3 is quite unsatisfying, as it is not referenced in the text and the results are not interpreted. Please elaborate on how the findings impact the results … what are the consequences of the results reported in Table 3? L392-439. It is not clear, why this section occurs in the discussion of the results – as it does not refer to the results? I would have expected it in the introduction and framing the study, i.e. determining characteristics for characterizing the studies found. In the way presented here, there seems to be no connection to the rest of the text. It has to made clearer, why this paragraph belongs to the rest of the text, either by discussion the results or by framing the study’s methodology.

Comments regarding formal issues

References: The title is included in quotation marks and concluded with a comma before the closing quotation mark. Looks a bit strange, please check if this is correct. Example: “The Emerging Field of Emotion Regulation: An Integrative Review*,*” L40-L41: “antecedent-focused strategies and response-focused to emotion regulation strategies” Sounds a bit strange. Please rephrase. L41: “However,“ does not fit. “Modifies” instead of “modify” L68 “It is known that through VR systems *it* is possible” L72 „the virtual world *it* is“ L76 “and clinicians to bring the person real-time live experiences in a safe environment” -> “… to expose patients to real-time ….” L87: The word “since” seems to be obsolete? L94-98 seem to lack conclusiveness: “Following the WHO, the purpose of the present review is to investigate whether wellbeing can be enhanced *using new technologies, such as virtual reality* [The present review is about VR only, so “such as is incorrect”?. Specifically, the present systematic review aims to better understand the efficacy of emotion regulation  interventions for wellbeing, by using virtual reality systems in adults, or older adults without  presenting psychopathological conditions. [I think this sentence is a better summary and can stand on its own, as both sentences tell the same using different words? – What do I have missed?]” L113: Caption of table 1 is placed too far away from table 1 and is IMHO misleading: This is not the strategy, but the results. Subsection 2.2 is the first subsection in section 2 -> Section 2.1 is missing. L126-L139: The logical flow of this section is not given. For example, the first sentence – which kind of study designs have been considered is not the most important characteristic of the studies considered. More important is the technical content, however this is described in the middle of the section. L149: “Third author”? Is this the third reviewer? Only two reviewers judged about each article, and if there was not a consensus, the third reviewer has been asked? Please clarify. L162: Subsection 3.3? Should be 3.1 L170: Wrong subsection number L178: Table Abbreviations EG and CG not introduced, Cell “Primary Outcomes” for article 10: “We …” -> Is this a literal quotation from the original paper? Then it should marked accordingly, otherwise please adjust to the reporting style of the other papers. I suggest trying to allocate the space better: e.g. width of Column “author” can be reduced, Column “age” can be reduced as well by abbreviating “Range” and “mean”, etc. L184: typo “Blindind” L572-572: Unbalanced Names in the authors list: “ Schek, Esther Judith Mantovani, O. Realdon, J. Dias, A. Paiva, and R. Schramm-Yavin, Sarit Pat-Horenczyk” some first names are abbreviated, some are not. L309-311: Empty lines? Consistent formatting regarding spacing between paragraphs and headings required. “technical interventions such as VR” might not sound so good. Either you list some other technical interventions, or you just call them VR-based interventions.

Author Response

We are very grateful to you for the time and involvement you have shown reviewing the manuscript. As you see, we revised the manuscript and incorporated the comments and recommendations. We have highlighted the text in all the revised parts in our manuscript from the original submission to facilitate the review process.

The comments we received were unpacked below in several points to facilitate a fluid understanding of the changes made. We hope that the changes improved the quality of the original manuscript considerably. Below we listed your detailed comments (italics) and our responses (red).

“There is still room for improvement in some aspects. One of the most important is the need to improve the logical structure. This applies both at chapter level and within the individual sections. At chapter level, the information developed in the introduction should be incorporated into the study. For example, the work of J.J. Gross is presented with good reason. But the influence on the design of the study, e.g. by defining the surveyed characteristics, does not really become clear. For example, what influence does the prominently presented distinction between antecedent-focused strategies and response-focused strategies have on the design of the study? Even in the discussion, topics are dealt with that do not occur in the actual study in this way. The discussion should be primarily based on the results. In its current form, the actual study appears to be a somewhat foreign body between introduction and discussion and some parts of the discussion appear superfluous.”

L392-439. It is not clear, why this section occurs in the discussion of the results – as it does not refer to the results? I would have expected it in the introduction and framing the study, i.e. determining characteristics for characterizing the studies found. In the way presented here, there seems to be no connection to the rest of the text. It has to made clearer, why this paragraph belongs to the rest of the text, either by discussion the results or by framing the study’s methodology.

Thank you for this observation.  Please see the changes made, according to your comment, on the logical structure of the paper for improving the readability in the revised version of ms. Following your comments, we changed the flow of contents. In the Introduction, we better explained the ER theoretical framework (line 36-58), without going into details of antecedent-focused strategies and response-focused strategies. In the previous version of the article, we focused on the specific components of Gross’s Model. The intention was to present a clear and scientifically relevant model, but the way it was dealt with was confusing and not useful in pursuing the core objectives of the ms.  

As regards the Result and Discussion sections, the structure was modified to make the manuscript more readable. Thanks to your observations, we had the opportunity to improve the manuscript, hoping it could be clearer, and more accessible to readers. To that end, we directed the revisions at  connecting the rationale presented in the first section throughout the following ones. The structure of the discussion was changed in order to highlight the results that emerged in the previous section. With the aim to facilitate understanding of the link between the various sections, the discussion was modified by ordering topics in the same order as the previous parts. In particular, from line 443 to 520 we discussed the studies with reference to the theoretical framework, following the logical structure of the introduction "From General to Specific".  

“Within the sections, the structure should also be critically reviewed. Examples are paragraphs 63-74 (ER and VR) and 75-91 (VR and intervention in psychology). In the current order they lead to duplication, especially when the second paragraph talks about wellbeing. In the reverse order, the principle "From General to Specific" would be respected.”

Thank you for your significant suggestion for improving the readability of our ms. As suggested, we proceeded to modify the order of the paragraphs indicated (line 60-80). We have also taken steps to make the content more streamlined and flowing, by removing redundant parts.

“In the current form of the paper I suggest to describe the content of each chapter in the beginning of the chapter as an orientation for the reader (and follow this description in the chapter strictly).”

“Section 3 lacks a systematic approach to evaluating the results, as the approach followed is hard to understand, it’s a kind of sequenced information. Alternatively, the approach followed should be better motivated using an introductory section. E.g., it does not become really clear, why a distinction is made between older adults and adults? Ok, there may be good reasons, and these reasons may have been explained in the introduction, but short explanations and introductions would be beneficial for this sections (IMHO).”

“E.g., section 3.7.2 could be enriched by a table showing the measurement instruments used and their characteristics. I suggest rewriting this section using a more systematic approach. For example, IMHO, the most important characteristic of VR experiences are the objectives pursued, and I would suggest presenting the findings of the study in an own, first section. Further, an alignment of the studies found with the Gross-model would be beneficial, as the model is mentioned in the introduction and in the discussion, but not in between.”

These are very meaningful observations for improving the readability of our ms. First of all, we added in each chapter an introduction of the following content to support a better orientation while reading. Concerning section 3, we introduced a new paragraph “Age differences in emotional experience” at the beginning of the section (line 194-208). In such a paragraph, we explained the reasons which brought us to split the analyses of VR-based ER interventions by the age of the sample. Furthermore, a more specific overview of the characteristics of interventions was added (specifically the VR characteristics, as the degree of immersion, the VE content, just as the description of the scenario, and the functional aim of each evaluated study). These contents regarding adults and older adults are presented in Table 4 (line 264) and Table 5 (line 301) respectively. We also followed your suggestion on the “3.7.2. section” about wellbeing measures. In fact, we added Table 6 (368) in which we explained the main characteristics of the scales. Contents in this section have been reworked; even if the reference to the theoretical framework of ER remains present only in the Discussion. This is due to the fact that we followed the PRISMA Guidelines, and therefore in the results we could make only an accurate and uncritical description of the evaluated studies.

The addition of tables helped the understanding of the investigated studies more usable. We believe that these additions were beneficial for the Result section.

“In some places, a more concrete writing style would be required, e.g. in the literature review of the introduction naming of the authers and some of their concrete results is IMHO helpful to improve the flow of the text.”

“A proofreading is strongly recommended. There are problems e.g. with article / no article, coherence of numerus of the subject and the verb with/without s. Only for section 1, formal issues are mentioned in the review, thereafter commenting formal issues has been stopped.”

We are aware of it, and we entirely agree with your observation. We have carefully checked the English language.

“At the moment, in the paper shows almost no striking results, which invite readers to cite the paper. The paper is a nice read but it is hard remembering concrete results. This could be improved by adding some tables with statistics (e.g. a categorization of the objectives of VR interventions, percentages of VR environment types (e.g. immersive and non-immersive) etc.”

Thank you for this observation. It is an exciting way to better organize our ms. Following your comments, we added Tables 4 and 5 in the results sections, as explained above. In these tables, we categorized the studies regarding their objectives and we presented VR types and the VE description of the scenarios. Then, at the beginning of the discussion (line 377-387), we reported the statistics (obtained) in the tables.

Remarks regarding the content. “L109-L110: IMHO, the search string lacks parentheses: "virtual reality" OR "virtual environment*" OR "digital intervention*" OR "digital technologies” AND "emotion regulation" OR "affect regulation" AND "wellbeing.". Precedence of the AND operator is higher than the precedence of the OR-Operator, so that the string evaluates to: "virtual reality" OR "virtual environment*" OR "digital intervention*" OR  ("digital technologies” AND "emotion regulation") OR ("affect regulation" AND "wellbeing.") which is not the intention of the authors. BTW: The full stop within the quotation marks is syntactically wrong. According to Table 1, the search string was: ("virtual reality" OR "virtual environment*" OR "digital intervention*" OR  "digital technologies”) AND ("emotion regulation" OR "affect regulation" *OR* "wellbeing."), i.e. the “and” before wellbeing must be an “or”?

This is a very meaningful observation. We followed your suggestion and we modified the search string according to the one presented here.

L119, L123: What is the difference between VR device and VR equipment – my impression is that there is no difference? Please use a consistent wording. Further, device is in my ears connotated with hardware. However, hardware develops fast and therefore is not that relevant IMHO. Ok, probably it should point to the difference between immersive and not immersive VR devices, but IMHO the expressions chosen are misleading.

More important are the VR applications or VR experiences, which have to be characterized in the review IMHO.

Thank you for allowing us to improve the fluency in this part. We are aware that an appropriate use of the terminology is crucial to avoid confusion. Hence, we have replaced the form of the sentences. During the proofreading phase, we rewrote some phrases using more consistent wording.

L166, Figure 1: First box: Please explain? Why is there only the database PubMed referenced? What is the difference between citation and article? I suggest relocating this figure into the vicinity of Table 1, because it explains parts of the methodology. Table 3 is quite unsatisfying, as it is not referenced in the text and the results are not interpreted. Please elaborate on how the findings impact the results … what are the consequences of the results reported in Table 3?

Thank you for these observations. These are mistakes that we had missed. We replaced the figure with the correct one and we reworded the content as you suggested. In order to facilitate the understanding of the process, we explained the figure at the beginning of the paragraph (158-160) suggesting to refer also to Table 1. Unfortunately, Table 1 and Figure 1 cannot be closer because the PRISMA Guidelines suggest to insert them in “2 Methods” and “3 Results” respectively.

As regards Table 3, we added an introduction of the paragraph that explains the contents of the table.

Comments regarding formal issues. References: The title is included in quotation marks and concluded with a comma before the closing quotation mark. Looks a bit strange, please check if this is correct. Example: “The Emerging Field of Emotion Regulation: An Integrative Review*,*” L40-L41: “antecedent-focused strategies and response-focused to emotion regulation strategies” Sounds a bit strange. Please rephrase. L41: “However,“ does not fit. “Modifies” instead of “modify” L68 “It is known that through VR systems *it* is possible” L72 „the virtual world *it* is“ L76 “and clinicians to bring the person real-time live experiences in a safe environment” -> “… to expose patients to real-time ….” L87: The word “since” seems to be obsolete? L94-98 seem to lack conclusiveness: “Following the WHO, the purpose of the present review is to investigate whether wellbeing can be enhanced *using new technologies, such as virtual reality* [The present review is about VR only, so “such as is incorrect”?. Specifically, the present systematic review aims to better understand the efficacy of emotion regulation  interventions for wellbeing, by using virtual reality systems in adults, or older adults without  presenting psychopathological conditions. [I think this sentence is a better summary and can stand on its own, as both sentences tell the same using different words? – What do I have missed?]” L184: typo “Blindind” L572-572: Unbalanced Names in the authors list: “ Schek, Esther Judith Mantovani, O. Realdon, J. Dias, A. Paiva, and R. Schramm-Yavin, Sarit Pat-Horenczyk” some first names are abbreviated, some are not. L309-311: Empty lines? Consistent formatting regarding spacing between paragraphs and headings required. “technical interventions such as VR” might not sound so good. Either you list some other technical interventions, or you just call them VR-based interventions.  L113: Caption of table 1 is placed too far away from table 1 and is IMHO misleading: This is not the strategy, but the results. Subsection 2.2 is the first subsection in section 2 -> Section 2.1 is missing. L126-L139: The logical flow of this section is not given. For example, the first sentence – which kind of study designs have been considered is not the most important characteristic of the studies considered. More important is the technical content, however this is described in the middle of the section. L149: “Third author”? Is this the third reviewer? Only two reviewers judged about each article, and if there was not a consensus, the third reviewer has been asked? Please clarify. L162: Subsection 3.3? Should be 3.1 L170: Wrong subsection number L178: Table Abbreviations EG and CG not introduced, Cell “Primary Outcomes” for article 10: “We …” -> Is this a literal quotation from the original paper? Then it should marked accordingly, otherwise please adjust to the reporting style of the other papers. I suggest trying to allocate the space better: e.g. width of Column “author” can be reduced, Column “age” can be reduced as well by abbreviating “Range” and “mean”, etc.  

Thank you for these critical suggestions for improving the readability and correctness of the English of our ms.  Please see the changes made according to your comment in the revised version of ms.

Thank you for your contribution to our work, we feel that thanks to this we have improved its value. We much appreciate that you found it interesting.

Round 2

Reviewer 1 Report

I am really impressive about revised version of the article.

Structure of the article is well reorganized and readability is quite improved.

Author Response

We are very grateful to you for the involvement you have shown in our work. The final version is better organized and readable thanks to your comments. The quality of our manuscript is improved also as a result of your revisions.

Reviewer 2 Report

In the present version, structure and layout have been changed a lot, a fluent reading is promoted. Many thanks to the authors for their efforts!

Some possible formal issues should still be checked:

L50 51 Please check: The emotion regulation process is a mechanism *that* may be used to make things either better or worse, depending on the context

L68 promising*,* because

L130 “we obtained 530 articles, of which 414 excluding duplicates.” Please check the phrase.

Figure 1: In the first row, there is still “Citations” instead of “Articles”

Author Response

We much appreciate the time you have spent on our manuscript. In our opinion, the new version of the article has gained added value as a result of your comments. As you see, we have incorporated the latest comments and recommendations.

Below we have reported your detailed comments (italics) and our responses (red).

Some possible formal issues should still be checked:

L50 51 Please check: The emotion regulation process is a mechanism *that* may be used to make things either better or worse, depending on the context

L68 promising*,* because

Thank you for this observation. We followed the suggestion and the modified text is visible in the final version of the article.

L130 “we obtained 530 articles, of which 414 excluding duplicates.” Please check the phrase.

Thank you for allowing us to improve the fluency in this part. We rewrote the phrase as follows “The initial searches on the databases yielded 530 results. Duplicates were removed leaving 414 articles for further evaluation.”

Figure 1: In the first row, there is still <<Citations>> instead of <<Articles>>”

Thank you for the observation. These are mistakes that escaped previously. We had previously replaced the figure as requested, but we had not noted these mistakes. We replaced the Figure with a new one made in ppt, and not with the automatic generator online as before.

 Thank you for your contribution to our work, we feel that thanks to this we have improved its quality.